# Ultrafast entropy production in pump-probe experiments

Lorenzo Caprini [1] ✉, Hartmut Löwen[1] & R. Matthias Geilhufe [2] ✉

The ultrafast control of materials has opened the possibility to investigate non-equilibrium states of matter with striking properties, such as transient super-conductivity and ferroelectricity, ultrafast magnetization and demagnetization, as well as Floquet engineering. The characterization of the ultrafast thermodynamic properties within the material is key for their control and design. Here, we develop the ultrafast stochastic thermodynamics for laser-excited phonons. We calculate the entropy production and heat absorbed from experimental data for single phonon modes of driven materials from time-resolved X-ray scattering experiments where the crystal is excited by a laser pulse. The spectral entropy production is calculated for $SrTiO_3$ and $KTaO_3$ for different temperatures and reveals a striking relation with the power spectrum of the displacement-displacement correlation function by inducing a broad peak beside the eigenmode-resonance.

Entropy production has been introduced in the nineteenth century to describe the amount of irreversibility in thermodynamic cycles. It is behind the formulation of the Clausius inequality and the second law of thermodynamics. More generally, it characterizes heat and mass transfer processes at the macroscopic scales[1], such as heat exchange, fluid flow, or mixing of chemical species. Furthermore, in terms of information-entropy, it plays a significant role in information theory[2].

Successively, entropy production has been linked to microscopic dynamics[3] to quantify the amount of irreversibility and dissipation at the atomistic (single-particle) level[4,5]. In the framework of gases, soft materials, or living organisms, each microscopic particle evolves in the presence of stochastic forces. These forces are usually generated by internal mechanisms, e.g., metabolic processes, internal motors, or collisions due to solvent molecules. The stochastic nature of the dynamics allows us to characterize macroscopic observables as averages of fluctuating variables, by considering the probability of observing a path of the microscopic trajectory. This approach is at the basis of stochastic thermodynamics[3], which aims of building the thermodynamic laws in terms of fluctuating work, heat, and entropy which on average are consistent with macroscopic thermodynamics[6].

In ordered phases of matter, we argue that thermal fluctuations of, e.g., ionic positions, spins, or charge lead to stochastic forces on microscopic degrees of freedom. Entropy is produced in non-equilibrium regimes, by excitations of the material with an external drive. This is motivated by immense progress in ultrafast control and characterization of crystalline solids[7-19]. We put specific focus on light-induced phonon dynamics[20-31]. Here, selected phonon modes are excited by strong THz laser pulses[32,33]. Remarkably, the ionic dynamics can be resolved with high precision with time-resolved X-ray scattering present at coherent X-ray light sources[34-46]. We deduce that the information obtained from such a scattering experiment is sufficient to reproduce the spectral entropy production rate of the medium within the material, giving rise to information about the ultrafast heat absorbed by the system.

In addition, characterizing and controling materials in terms of thermal properties in the ultrafast regime has emerged as a powerful research path[15,47]. Hence, developing stochastic thermodynamics properties generated at short time scales, e.g., entropy production and heat, could open new perspectives for the comprehension of functional materials. In the following, we show that non-equilibrium crystals, driven by a laser pulse, are characterized by spectral entropy production. As illustrated in Fig. 1, we propose to measure entropy production of the medium from ionic displacements, e.g., obtained from time-resolved X-ray scattering experiments. Further, we show that the power spectrum of ionic displacement shows a close connection to the spectral entropy production. We compare our theory to

[1]Institut für Theoretische Physik II: Weiche Materie, Heinrich-Heine-Universität Düsseldorf, 40225 Düsseldorf, Germany. [2]Department of Physics, Chalmers University of Technology, 412 96 Göteborg, Sweden. ✉e-mail: lorenzo.caprini@gssi.it; matthias.geilhufe@chalmers.se

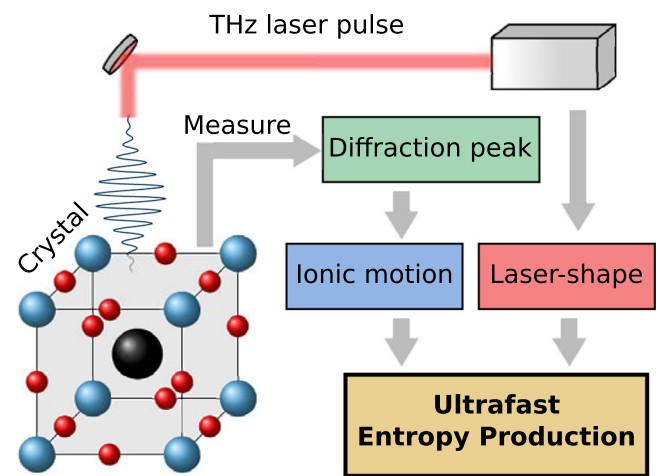

**THz laser pulse**

Measure

Diffraction peak

Crystal

Ionic motion    Laser-shape

**Ultrafast Entropy Production**

**Fig. 1 | Schematic representation of a crystal (SrTiO₃ or KTaO₃) excited by a THz laser pulse.** From a direct measure of the diffraction pattern, for instance, obtained from time-resolved X-ray scattering experiments, the ionic displacement can be deduced. Combining this measure with the shape of the THz laser pulse, we can calculate the ultrafast entropy production of the medium by applying our theoretical results.

experimental data for SrTiO₃ and support our approach by providing estimates for the soft modes of KTaO₃ and SrTiO₃.

## Results

### Ultrafast stochastic thermodynamics of crystals
We model the dynamics of an optical phonon mode by the equation of motion[48–56]

$$\ddot{u}(t) + \eta \dot{u}(t) + \omega_0^2 u(t) = \sqrt{2\eta\,k_B T}\,\xi(t) + F(t), \tag{1}$$

Here, $k_B$ is the Boltzmann constant while $u(t)$ is a phonon normal mode (units $\text{Å}\sqrt{\text{a.m.u}}$) with frequency $\omega_0$, and damping or line width $\eta$. $F(t)$ is an external driving field, which, for a laser excitation can be written as $F(t) = Z\tilde{E}(t)$. $Z$ is the mode effective charge[57], $\tilde{E}(t) = \epsilon^{-1}E(t)$ the screened electric field, and $\epsilon$ the relative permittivity.

For simplicity, we neglect nonlinear effects[22,55,56,58]. Furthermore, we add an uncorrelated noise $\sqrt{2\eta\,k_B T}\,\xi(t)$ which models the interaction of the phonon normal mode with thermally excited lattice fluctuations $\xi$ at the environmental temperature $T$[59,60]. The equation of motion (1) has a formal solution in Fourier space, given by

$$\hat{u}(\omega) = \chi(\omega)\left(\sqrt{2\eta\,k_B T}\,\hat{\xi}(\omega) + \hat{F}(\omega)\right), \tag{2}$$

with the susceptibility $\chi(\omega) = \left(\omega_0^2 - \omega^2 + i\eta\omega\right)^{-1}$. An example of the solution in real-time is reported in the methods section. Let $u = \{u\}$ denote a specific solution or trajectory between the initial time $t_0$ and the final time $\mathcal{T}$, with the initial conditions $u_0$. The presence of thermal noise in the equation of motion introduces a final probability of realizing $\{u\}$, given by $P[\{u\}|u_0]$. The force $F(t)$ breaks the time-reversal symmetry. As a consequence, the probability of observing the time-reversed path $P_r[\{u\}|u_0]$ differs from $P[\{u\}|u_0]$[61–63]. This generates entropy production of the medium, $\Sigma(t)$,

$$\Sigma(t) = k_B \log\frac{P[\{u\}|u_0]}{P_r[\{u\}|u_0]} = \int_0^t d\tau\,\dot{s}(\tau), \tag{3}$$

where we have conveniently introduced $\dot{s}(t)$ as the entropy production rate of the medium. This observable can be naturally identified as the stochastic heat flow absorbed by the system divided by temperature[64,65]. In the case of uncorrelated noise $\langle\xi(t)\xi(t')\rangle \sim \delta(t - t')$,

the entropy production rate of the medium (Eq. (3)) is given by $\dot{s}(t) = \langle v(t)F(t)\rangle/T$, with $v(t) = \dot{u}(t)$[3,62]. Note, that this relation is general and thus also holds for non-linear phonon dynamics. By decomposing $\Sigma$ in Fourier waves[66,67], we introduce the spectral entropy production of the medium $\hat{\sigma}(\omega)$ as

$$\hat{\sigma}(\omega) = \int d\omega'\,S_r(\omega,\omega'), \tag{4}$$

with the entropy spectral density

$$S_r(\omega,\omega') = \frac{i}{T}\omega'\chi(\omega')\hat{F}(\omega')\hat{F}(\omega - \omega'). \tag{5}$$

Equations (4) and (5) are central theoretical results of the paper. With the knowledge of the susceptibility and the shape of the applied drive, quantities typically accessible in experiments, the spectral entropy production of the medium, and thus the heat flow, can be determined (Fig. 1). As a result, our predictions hold beyond phonons and can be applied for other excitations. In stochastic systems, the entropy production rate is a real fluctuating observable but its time average is positive in agreement with the second law of thermodynamics. In contrast, spectral entropy production is generally complex. To shed light on the interpretation of the spectral entropy production of the medium $\hat{\sigma}(\omega)$, we note it can be evaluated analytically for a periodic driving field $F(t) = A\exp(i\omega_d t)$. The imaginary part of $\hat{\sigma}$ follows to be $\Im\hat{\sigma} = \delta(\omega - 2\omega_d)A^2(T)^{-1}\omega_d(\omega_0^2 - \omega_d^2)\left((\omega_0^2 - \omega_d^2)^2 + \eta^2\omega_d^2\right)^{-1}$. Hence, it shows a delta peak at twice the driving frequency $\omega_d$. Furthermore, it is negative (positive) if the driving frequency $\omega_d$ is larger (smaller) than the eigenfrequency $\omega_0$. In contrast, the real part $\Re\hat{\sigma} = \delta(\omega - 2\omega_d)A^2(T)^{-1}\omega_d^2\eta\left((\omega_0^2 - \omega_d^2)^2 + \eta^2\omega_d^2\right)^{-1}$ is an odd function of the damping $\eta$. Therefore, $\Re\hat{\sigma}$ vanishes for zero damping. Hence, $\Re\hat{\sigma}$ is a measure of dissipation associated with $\eta$. Both, $\Im\hat{\sigma}$ and $\Re\hat{\sigma}$ decrease with the distance between eigenfrequency $\omega_0$ and driving frequency $\omega_d$ as well as with increasing temperature.

### The power spectrum and spectral entropy production
The spectral entropy production of the medium can be determined from the frequency profile of the external force, e.g., THz laser pulses, and the susceptibility of the system. Alternatively, the power spectrum $\langle u(t)^2\rangle$ can be expressed in terms of the entropy production generated by the laser excitation and, therefore, can be used to extract ultrafast thermodyamics properties of the system. Evaluating $\langle u(t)^2\rangle$ in Fourier space, the power spectrum can be decomposed in two contributions as (see detail in the methods section)

$$\mathcal{F}\left[\langle u(t)^2\rangle\right](\omega) = \mathcal{F}\left[\langle u(t)^2\rangle\right]_{\text{eq}}(\omega) + \mathcal{F}\left[\langle u(t)^2\rangle\right]_{\text{neq}}(\omega). \tag{6}$$

The first one $\mathcal{F}[\langle u(t)^2\rangle]_{\text{eq}}$ has an equilibrium origin and, indeed, arises from thermal fluctuations,

$$\mathcal{F}\left[\langle u(t)^2\rangle\right]_{\text{eq}} = 2\eta\,k_B T\delta(\omega)\int\frac{d\omega'}{2\pi}\hat{\chi}(\omega')\hat{\chi}(-\omega'). \tag{7}$$

As a result, it is $\propto T\delta(\omega)$ and fully determined by the susceptibility $\chi$.

In contrast, the term $\mathcal{F}[\langle u(t)^2\rangle]_{\text{neq}}$ originates from the external field (THz laser pulse) and, thus, reflects the non-equilibrium part of the dynamics. Indeed, this term (see "Methods" section) can be

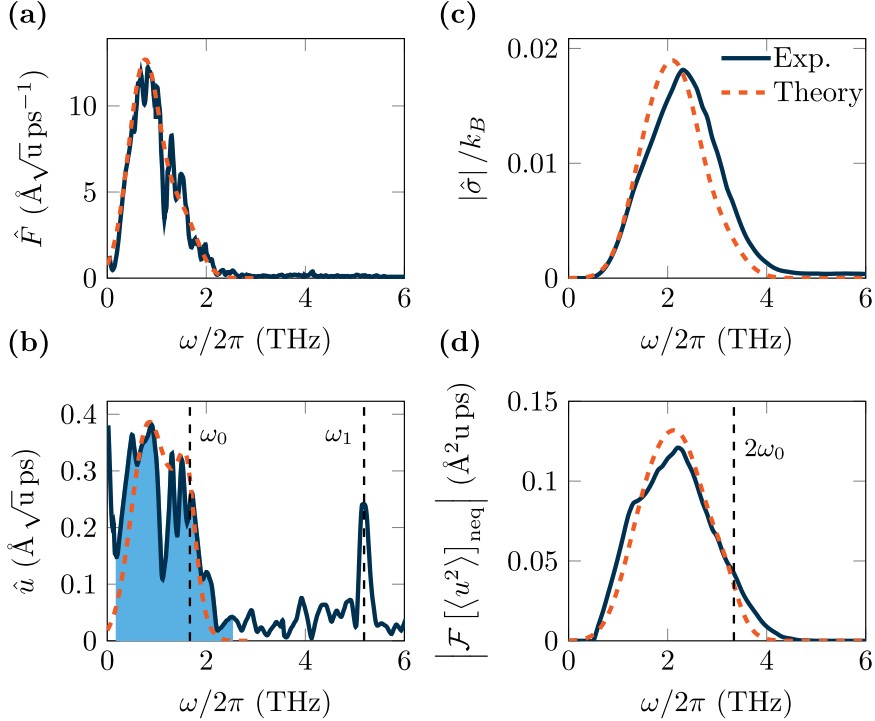

**Fig. 2 | Entropy production of the medium in SrTiO$_3$ after exposure to an intense THz laser pulse at 100 K.** We compare an estimate computed from time-resolved X-ray scattering data taken from Kozina et al.[22] with model data. **a** Fourier transform of the THz laser pulse (solid dark blue), compared with a theoretical Gaussian laser pulse (orange dashed) with frequency $\omega_d = 0.75$ THz, superposed with a higher-harmonic at $2\omega_d$. **b** Comparison of experimental (solid dark blue) and computed (dashed orange) Fourier transform of the phonon normal mode amplitude, $\hat{u}(\omega)$. The soft mode contribution is shaded in light blue. **c** Comparison of the spectral entropy production of the medium, $|\tilde{\sigma}|$, computed from the full experimental data of the phonon normal mode amplitude (solid dark blue) with our model taking into account the soft mode only (dashed orange). **d** Comparison of the power spectrum, $|\mathcal{F}[\langle u^2 \rangle]_{\text{neq}}|$, computed from the full experimental data (solid dark blue) with our soft-mode-only model (dashed orange).

expressed in terms of the entropy spectral density, $S_r(\omega, \omega')$, and reads

$$\mathcal{F}_\omega \langle u^2(t) \rangle_{\text{neq}} = T \int \frac{d\omega'}{2\pi} \frac{\hat{\chi}(\omega - \omega')}{(i\omega')} \hat{S}_r(\omega, \omega'). \tag{8}$$

Relation (8) is a key result of the paper providing an alternative route to measure the spectral entropy production of the medium, e.g., heat flow. It shows that the ultrafast spectral entropy production in crystals can be measured from the power spectrum of the phonon displacement, an observable signature. In particular, $\langle u^2(t) \rangle$ is measured in time-resolved diffuse X-ray scattering[68,69].

**Application to SrTiO$_3$ and KTaO$_3$ under laser pulses**
To show that heat, i.e., entropy production rate of the medium multiplied by the environmental temperature, can be obtained from experiments, we compare our model to time-resolved X-ray scattering data obtained by Kozina et al., for the nonlinear excitation of phonons in SrTiO$_3$[22]. The spectral components of the used THz laser pulse are shown in Fig. 2a. To sufficiently reproduce the shape of the spectrum, we assume a superposition of two Gaussian laser pulses, one at frequency $\omega_d = 0.75$ THz and a higher-harmonic component with $2\omega_d$, $F(t) = Z\tilde{E}_0 \left( \exp(2\pi i \omega_d t) + \alpha \exp(4\pi i \omega_d t) \right) \exp(-\frac{1}{2}\frac{t^2}{\tau^2})$, with $\alpha \approx 0.2858$. The in-medium field strength is $\beta E_0$, with $\beta = 0.215$ and $E_0 = 480$ kV cm$^{-1}$, while the pulse width is $\tau = 0.5$ps. The experiment was performed at 100 K, where the soft mode frequency is measured to be $\omega_0/2\pi \approx 1.669$ THz with a damping of $\eta/2\pi \approx 0.9$ THz. The mode effective charge of SrTiO$_3$ is $Z = 2.6$ e$^-$ a.m.u.$^{-1/2}$ [22,51], with e$^-$ the elementary charge and u.m.u. the atomic mass unit.

The measured spectral component of the time-domain X-ray data[22] is scaled against the computed amplitude of the soft mode according to Eq. (2) and shown in Fig. 2b. The soft mode contribution

to the experimental spectrum is shaded in light blue. Data are used to compute the spectral entropy production, $|\tilde{\sigma}|$, of the soft mode as a function of $\omega$ and compared against our theory in Fig. 2c. $|\tilde{\sigma}|$ computed from the experimental data exhibits a peak at frequency $\omega_{\sigma_1}/2\pi \approx 2.33$ THz, that is reproduced by our model. Furthermore, we reconstruct the power spectrum, $|\mathcal{F}[\langle u^2 \rangle]_{\text{neq}}|$, given in Fig. 2d, which is off-resonance with twice the soft-mode frequency. The shape of $|\mathcal{F}[\langle u^2 \rangle]_{\text{neq}}|$ shows strong overlap with the computed entropy production. Due to nonlinear coupling between phonons, discussed in ref. 22, a peak of the second optical mode at $\approx 5.19$ THz can be clearly observed in Fig. 2b. We note that this mode (not considered in our model) has no spectral overlap with the driving field, which is almost zero for $\omega > 3$THz. As a result, the entropy production generated by the second optical mode and the laser field is negligible (compare Fig. 2b and c, see also "Methods" section).

To shed light on heat induced by laser fields, we compute the spectral entropy production of the medium for two different materials SrTiO$_3$ and KTaO$_3$. Here, in contrast to the previous assumptions, we consider a single Gaussian laser pulse, $F(t) \sim e^{2\pi i \omega_d t}$, for simplicity, without higher-harmonic contribution. We fix the in-medium field strength to be $\tilde{E}_0 = 100$ kV cm$^{-1}$. As before, the frequency of the driving field is $\omega_d = 0.75$THz and the pulse width is $\tau = 1$ps. The mode effective charges are, SrTiO$_3$: $Z = 2.5$[22,51], KTaO$_3$: $Z = 1.4$[49,51]. We focus on the soft mode for which phonon frequency $\omega_0 = \omega_0(T)$ and line width $\eta = \eta(T)$ strongly depend on temperature[70,71] (see "Methods" section).

SrTiO$_3$ is a cubic perovskite with a tetragonal phase transition at $\approx 105$K[72]. Further, SrTiO$_3$ exhibits a diverging dielectric constant at low temperatures as well as an asymptotic vanishing of the soft-mode frequency, both indicative of a ferroelectric phase transition[71,73]. However, the transition is avoided due to quantum fluctuations, making SrTiO$_3$ a quantum critical paraelectric[73]. According to ref. 70,

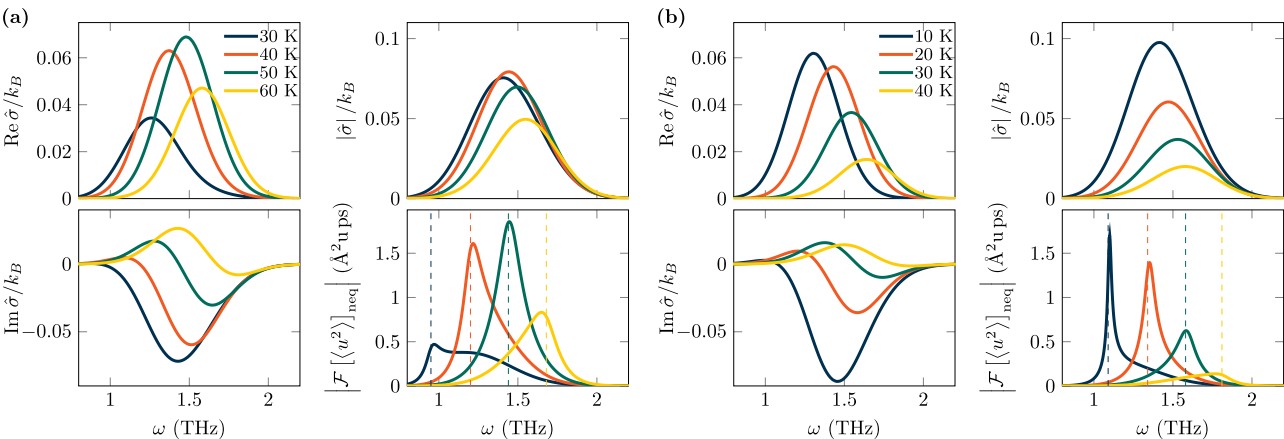

**Fig. 3 | Ultrafast spectral entropy production.** Results are shown for SrTiO₃ (**a**) and KTaO₃ (**b**). Each color refers to a different temperature $T$. In each case, real part Re $\hat{\sigma}(\omega)$, imaginary part Im $\hat{\sigma}(\omega)$ and modulus $|\hat{\sigma}(\omega)|$ of the spectral entropy production of the medium, $\hat{\sigma}(\omega)$, (see definition (4)), are shown together with the

Fourier transform of the non-equilibrium contribution of the power spectrum $\mathcal{F}_\omega \langle u^2(t) \rangle_{\mathrm{neq}}$. Temperature-dependent soft modes are considered (see methods section). Dashed lines in the plot for $\mathcal{F}_\omega \langle u^2 \rangle_{\mathrm{neq}}$ denote twice the soft-mode frequency.

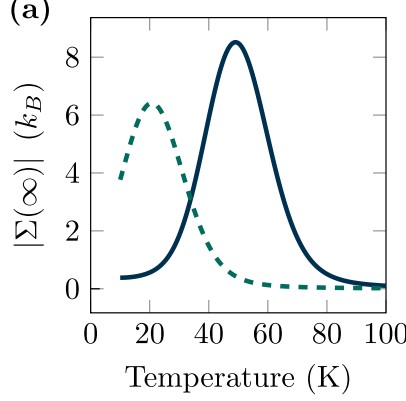

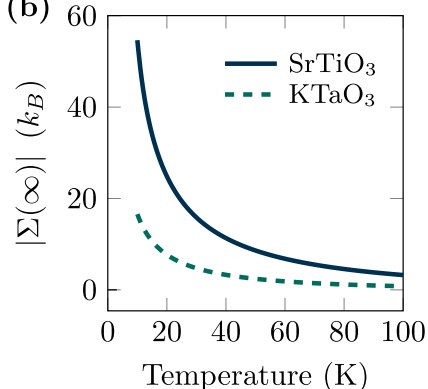

**Fig. 4 | Total entropy production of the medium.** Solid lines denote SrTiO₃, while dashed lines KTaO₃. The total entropy production of the medium, $|\Sigma(\infty)|$, is plotted as a function of temperature $T$, after the laser pulse has fully decayed. **a** The driving frequency is fixed to $\omega_d = 0.75$THz. Resonance peaks emerge at 52 K (SrTiO₃) and

26.4 K (KTaO₃) due to the temperature dependence of the soft mode. **b** The driving frequency is chosen to match the temperature-dependent soft mode frequency, $\omega_d = \omega_0(T)$.

the soft-mode frequency of SrTiO₃ is in resonance with the driving frequency, $\omega_0 = \omega_d = 0.75$THz at $T \approx 52$K. In Fig. 3a, we show the computed spectral entropy production of the medium for SrTiO₃ at various temperatures ranging from 30 to 60 K. Due to the temperature dependence and softening of the damping, the real part becomes maximal slightly above 50 K. In contrast, the imaginary part of $\hat{\sigma}(\omega)$ increases with decreasing temperature, showing a clear sign change below 52 K. The absolute value of the spectral entropy production shows a local maximum around this temperature. Due to the narrow width of the Gaussian laser field (1 ps), neither Re $\hat{\sigma}$, Im $\hat{\sigma}$, nor $|\hat{\sigma}|$ have a peak at exactly $2\omega_d$, but instead show a decreasing peak frequency with decreasing temperature. Plotting $\mathcal{F}_\omega \langle u^2 \rangle_{\mathrm{neq}}$ reveals clear peaks at twice the soft-mode frequency, which is indicated by dashed lines. A non-symmetric broadening of the peak for frequencies occurs in agreement with the spectral weight of the spectral entropy production of the medium $\hat{\sigma}$. This becomes specifically apparent for the temperatures 30K, 40K and 60K. Interestingly, the connection between a non-symmetric broadening and entropy production has recently been discussed for active crystals, i.e., periodic arrangements of self-propelled particles, such as bacteria, cells, or Janus colloids. In those cases, the basic constituents of the crystal produce entropy in contrast to the present paper where entropy is generated by an external laser

source. This has led to the concept of entropons as a collective signature for spectral entropy production[74].

In contrast to SrTiO₃, KTaO₃ remains cubic to liquid helium temperatures[75]. It is also regarded a quantum paraelectric, but outside the quantum critical regime[76]. As a result, the decrease of the soft-mode frequency and damping is slower compared to SrTiO₃, being in resonance with the driving frequency $\omega_d = 0.75$THz at $\approx 26.4$K[70]. Therefore, we evaluate the spectral entropy production for temperatures between 10,...,40 K, plotted in Fig. 3b. The steady increase of Re $\hat{\sigma}$ with decreasing temperature shows that the spectral entropy production process dominates the decrease of the soft-mode damping. As before, the sign change of Im $\hat{\sigma}$ for low temperatures can be clearly revealed. In agreement with the absence of a theoretical ferroelectric transition at low temperatures, the soft mode frequency remains finite at low temperatures. As a result, the soft-mode peaks at $2\omega_0$ in $\mathcal{F}_\omega \langle u^2 \rangle_{\mathrm{neq}}$ remain at higher frequencies, compared to SrTiO₃. Furthermore, the peaks occur fairly close to the maxima of $|\hat{\sigma}|$ making the entropon broadening less pronounced, in comparison to SrTiO₃.

Our theory allows us to estimate the total amount of dissipation due to the laser pulse, by calculating the total entropy production of the medium $\Sigma(\infty)$ according to Eq. (3). This observable is shown for SrTiO₃ and KTaO₃, in Fig. 4, where we have used a real-valued

driving force and computed the entropy production $\Sigma(\infty)$ from a direct solution of the equation of motion (1). For a fixed laser frequency, we observe that the total entropy production of the medium is maximized when the ferroelectric soft mode is in resonance with the driving frequency, i.e., at $T \approx 52K$ for $SrTiO_3$ and at $T \approx 26.4K$ for $KTaO_3$, respectively (Fig. 4a). As soon as the soft mode frequency is out of resonance, the entropy production of the medium is suppressed. This implies that a crystal is characterized by a non-monotonic capacity of absorbing heat and producing entropy when subject to a driving force. The maximal absorbed heat is a result of a resonant effect between phonon modes and driving frequencies. To characterize this resonant effect, we investigate its temperature dependence. Specifically, in Fig. 4b, we vary the driving frequency to match the soft mode frequency at each temperature by setting $\omega_d = \omega_0(T)$. The computed total entropy production of the medium increases with decreasing temperature and, in particular, diverges for $T \to 0$. As a result, in resonant conditions, the smaller the temperature the larger the maximal absorbed heat.

## Discussion

We studied ultrafast thermodynamic processes, by deriving the absorbed heat, e.g., the ultrafast entropy production of the medium, due to transient phonons in materials excited by a THz laser pulse. Specifically, the soft modes of $SrTiO_3$ and $KTaO_3$ are evaluated by comparing our theory to experimental data and simulation results. The entropy production of the medium takes place on the picosecond timescale and can be deduced from the collective ionic displacement as observed, e.g., by time-resolved X-ray scattering. While entropy production and sample heating are well-known concepts in general, our work sheds light on the microscopic mechanism behind entropy production in driven quantum materials, using the framework of stochastic thermodynamics. While the maximal energy transfer from the laser to the sample is determined by the laser intensity, the production of entropy strongly increases with decreasing temperature. Furthermore, the temporal signature of this process is tightly bound to the soft mode frequency.

More generally, we envision ultrafast thermodynamics to provide characteristic signatures of complex systems, beyond phononic processes. We showed, in particular, that, in the presence of uncorrelated noise, entropy production depends on the materials' response function. As a consequence, Eq. (5) can be straightforwardly applied to other collective excitations, as long as they are discussed in the linear regime. A particularly interesting extension of our theory would concern magnons[8,16,77]. However, since magnetic systems can be governed by correlated noise and non-Markovian dynamics[78], a generalization of our theory to include these effects is required.

Coupling between phononic and magnonic degrees of freedom represents another promising research line to apply our theory. On the one hand, it has been recently shown that circularly polarized or chiral phonons can induce significant magnetization in nominally non-magnetic crystals[21,48,79–81]. This feature becomes particularly interesting when such a transient magnetization is used to switch magnetic orders in layered structures[82]. On the other hand, influencing magnetization by ultrafast heat production has been investigated intensively[47]. In addition, in the case of the optically induced magnetization due to the inverse Faraday effect, the validity of a thermodynamic picture of magnetization has been strongly debated[83–85]. Hence, we believe that our theory can be insightful in these contexts.

Furthermore, our theory for the power spectrum of the displacement-displacement correlation exhibits spectral weight besides a sharp peak at twice the eigenfrequency of the soft mode. We have shown that this part of the power spectrum can be associated with spectral entropy production. The emergence of such a feature is closely related to the concept of entropons recently introduced for

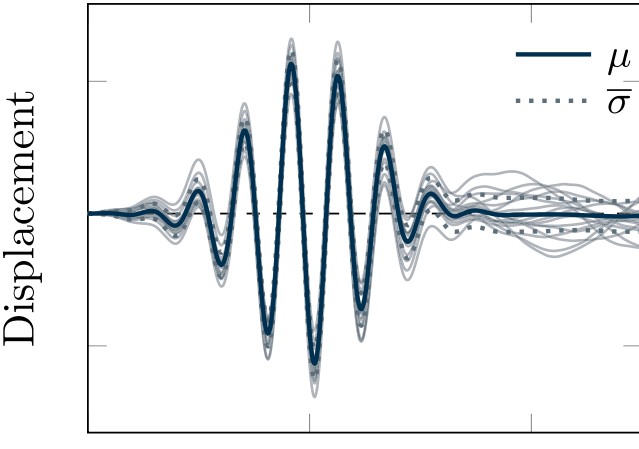

**Fig. 5 | Ensemble of solutions of the stochastic equations of motion.** The plot is obtained for uncorrelated noise and generic parameters. The blue solid line represents the mean solution, while dashed lines are single trajectories that illustrate the standard deviation.

intrinsic non-equilibrium systems reaching a steady-state[74]. In contrast, here, the system is away from the steady state, and entropy production is generated by the transient force due to laser pulses.

## Methods
### Spectral entropy production
By applying the time Fourier transform to the equation of motion for $u(t)$, Eq. (1) (see Fig. 5 for an example of its solution in real-time), we obtain the dynamics in the domain of frequency $\omega$

$$(-\omega^2 + \omega_0^2 + i\omega\eta)\hat{u}(\omega) = \sqrt{2\eta\,k_B T}\,\hat{\xi}(\omega) + \hat{F}(\omega), \qquad (9)$$

where the hat-symbol denotes the time-Fourier transform of a variable and $\hat{\xi}(\omega)$ is a Gaussian noise with zero average and $\langle \hat{\xi}(\omega)\hat{\xi}(\omega')\rangle = \delta(\omega + \omega')$. By defining the vector $v(t) = \dot{u}(t)$, so that $\hat{v}(\omega) = i\omega\hat{u}(\omega)$, Eq. (9) can be expressed as

$$i\omega\hat{u}(\omega) = \hat{v}(\omega) \qquad (10)$$

$$(i\omega + \eta)\hat{v}(\omega) + \omega_0^2\hat{u}(\omega) = \sqrt{2\eta\,k_B T}\,\hat{\xi}(\omega) + \hat{F}(\omega). \qquad (11)$$

The path-probability of the phonon normal mode, $P[\{u\}|u_0]$, conditioned to the initial value $u_0$, can be estimated by the probability distribution of the noise history $p[\{\xi\}|\xi_0]$, conditioned to the initial value $\xi_0$. Here, curly brackets denote the time history from the initial to the final time. The Gaussian properties of the noise allows us to express $p[\{\xi\}|\xi_0]$ as[63]

$$
\begin{aligned}
p[\{\xi\}|\xi_0] &\sim \exp\left(-\frac{1}{2}\int dt\,\xi(t)^2\right)\\
&= \exp\left(-\frac{1}{2}\int dt \int \frac{d\omega}{2\pi} e^{-i\omega t}\int ds\, e^{i\omega s}\xi(s)^2\right)\\
&= \exp\left(-\frac{1}{2}\int dt \int \frac{d\omega}{2\pi} e^{-i\omega t}\int \frac{d\omega'}{2\pi}\hat{\xi}(\omega')\hat{\xi}(\omega - \omega')\right),
\end{aligned}
\qquad (12)
$$

where in the second and third equalities we have applied the properties of Fourier transforms. From here, we can switch to the probability of the trajectory for the phonon mode $\{u\}$ by handling the change of

variables $\xi \to u$, i.e., by using the equation of motion in Fourier space

$$\hat{\xi}(\omega) = \frac{1}{\sqrt{2\eta k_B T}}[(i\omega + \eta)\hat{v}(\omega) + \omega_0^2 \hat{u}(\omega) - \hat{F}(\omega)]. \quad (13)$$

Such a change of variables should involve the determinant of the transformation. We ignore this term because it is irrelevant to the calculation of the entropy production since it provides only an even term under time-reversal transformation[63]. As a consequence, the following relation holds

$$P[\{u\}|u_0] \sim p[\{\xi\}|\xi_0]. \quad (14)$$

The path-probability of the backward trajectory of the phonon normal mode, $P_r[\{u\}|u_0]$, can be obtained by simply applying the time-reversal transformation (TRT) to the particle dynamics. By denoting time-reversed variables by a subscript $r$, the path-probability of the time-reversed noise history, $p_r[\{\xi\}|\xi_0]$, is still Gaussian and reads

$$p_r[\{\xi\}|\xi_0] \sim \exp\left(-\frac{1}{2}\int dt\, \xi_r(t)^2\right)$$
$$= \exp\left(-\frac{1}{2}\int dt \int \frac{d\omega}{2\pi} e^{-i\omega t} \int \frac{d\omega'}{2\pi} \hat{\xi}_r(\omega')\hat{\xi}_r(\omega - \omega')\right). \quad (15)$$

To switch to $P_r[\{\xi\}|\xi_0]$, we first have to evaluate the backward dynamics, by simply applying the TRT to Eq. (1). By using $u_r = u$ and $v_r = -v$, we conclude that all the terms in Eq. (1) are invariant under TRT except for the friction force. Applying the Fourier transform to Eq. (1) and expressing the noise $\hat{\xi}_r(\omega)$ as a function of $u_r(\omega)$ and $v_r(\omega)$, we can recur to the change of variable $\xi_r \to u$ that allows us to use the following relation

$$\hat{\xi}_r(\omega) = \frac{1}{\sqrt{2\eta k_B T}}\left[(i\omega - \eta)\hat{v}(\omega) + \omega_0^2 \hat{u}(\omega) - \hat{F}(\omega)\right]. \quad (16)$$

By neglecting again the determinant of the change of variables, $P_r[\{u\}|u_0]$ reads

$$P_r[\{u\}|u_0] \sim p_r[\{\xi\}|\xi_0]. \quad (17)$$

To calculate the entropy production $\Sigma$, we use the definition (3), i.e., the log-ratio between the probabilities of forward and backward trajectories of the phonon normal mode,

$$(2T)\Sigma = (2k_B T)\log\frac{p(\{u\}|u_0)}{p_r(\{u\}|u_0)}$$
$$= \int dt \int \frac{d\omega}{2\pi} e^{-i\omega t} \int \frac{d\omega'}{2\pi} \quad (18)$$
$$\times \left(\langle \hat{v}(\omega')\hat{F}(\omega - \omega')\rangle + \langle \hat{v}(\omega - \omega')\hat{F}(\omega')\rangle\right).$$

By comparing Eq. (18) with the definition

$$\Sigma = \int dt\, \dot{s}(t), \quad (19)$$

one can identify the entropy production rate, $\dot{s}(t)$, as

$$\dot{s}(t) = \int \frac{d\omega}{2\pi} e^{-i\omega t} \int \frac{d\omega'}{2\pi} \frac{1}{2T} \times \quad (20)$$

$$\times \left(\langle \hat{v}(\omega')\hat{F}(\omega - \omega')\rangle + \langle \hat{v}(\omega - \omega')\hat{F}(\omega')\rangle\right). \quad (21)$$

Applying the Fourier transform, we introduce the spectral entropy production rate, $\hat{\sigma}(\omega)$, as

$$\dot{s}(t) = \int \frac{d\omega}{2\pi} e^{-i\omega t}\hat{\sigma}(\omega) \quad (22)$$

and, by comparison with Eq. (20), we obtain

$$\hat{\sigma}(\omega) = \int \frac{d\omega'}{2\pi} \frac{1}{2k_B T} \left(\langle \hat{v}(\omega')\hat{F}(\omega - \omega')\rangle + \langle \hat{v}(\omega - \omega')\hat{F}(\omega')\rangle\right). \quad (23)$$

We remark that expressions (18) and (23) do not depend on the choice of the force in the dynamics of $\hat{u}(\omega)$. As a result, they are unchanged by adding a non-linear force, e.g., due to phonon-phonon coupling to Eq. (9).

Finally, we mention that the dissipative properties of a chain of harmonic oscillators have been previously studied with a stochastic thermodynamics approach[86,87]. In contrast, here, we focus on the entropy production associated to each collective excitations, e.g., phonons, by explicitly modeling the dynamics of an optical phonon excited by a THz laser pulse (Eq. (1)).

### Entropy spectral density

The formal solution of the equation of motion (1) in Fourier space is given by

$$\hat{u}(\omega) = \frac{\sqrt{2\eta\, k_B T}\, \hat{\xi}(\omega) + \hat{F}(\omega)}{\omega_0^2 - \omega^2 + i\omega\eta} = \chi(\omega)\hat{A}(\omega). \quad (24)$$

Here, $\chi(\omega)$ is the (linear) susceptibility

$$\chi(\omega) = \frac{1}{\omega_0^2 - \omega^2 + i\omega\eta}, \quad (25)$$

and $\hat{A}(\omega) = \sqrt{2\eta\, k_B T}\, \hat{\xi}(\omega) + \hat{F}(\omega)$. By using that $\hat{v}(\omega) = i\omega\hat{u}(\omega)$ and $\langle \hat{\xi}(\omega)\rangle$, the spectral entropy production, $\hat{\sigma}(\omega)$, can be expressed as

$$\hat{\sigma}(\omega) = \frac{i}{T}\int \frac{d\omega'}{2\pi} k\hat{F}(\omega - \omega')\chi(\omega')F(\omega'). \quad (26)$$

By introducing the entropy spectral density, $\hat{S}_r(\omega, \omega')$, as

$$\hat{\sigma}(\omega) = \int \frac{d\omega'}{2\pi} \hat{S}_r(\omega, \omega'), \quad (27)$$

we can immediately identify

$$\hat{S}_r(\omega, \omega') = \frac{(i\omega')}{T}\hat{F}(\omega - \omega')\chi(\omega')F(\omega'). \quad (28)$$

Equation (28) coincides with formula (5) of the main text. Non-linear force terms do not allow the system to have a formal solution in terms of $\chi(\omega)$. Thus, formula (28) holds only in the linear case.

### Dynamical correlation of the normal phonon mode

By using Eq. (24) the Fourier transform of the dynamical correlation, $\mathcal{F}\langle u^2(t)\rangle$, is given by

$$\mathcal{F}\langle u^2(t)\rangle = \int \frac{d\omega'}{2\pi} \langle \hat{u}(\omega')\hat{u}(\omega - \omega')\rangle$$
$$= \int \frac{d\omega'}{2\pi} \langle \hat{A}(\omega')\hat{A}(\omega - \omega')\rangle \hat{\chi}(\omega')\hat{\chi}(\omega - \omega'). \quad (29)$$

First, we applied the convolution theorem and, second, we used Eq. (24). Using the definition of $\hat{A}(\omega)$, $\mathcal{F}_\omega\langle u^2(t)\rangle$ can be decomposed

into two terms,

$$\mathcal{F}_\omega \langle u^2(t)\rangle = \mathcal{F}_\omega \langle u^2(t)\rangle_{eq} + \mathcal{F}_\omega \langle u^2(t)\rangle_{neq}. \tag{30}$$

The first term, $\mathcal{F}_\omega \langle u^2(t)\rangle_{eq}$, in the right-hand side of Eq. (29), has an equilibrium origin: it arises from the Brownian noise and is given by the convolution of the susceptibility with itself. For uncorrelated noise, we have $\langle \hat{\xi}(\omega')\hat{\xi}(\omega - \omega')\rangle = \delta(\omega)$, and this term reads

$$\mathcal{F}_\omega \langle u^2(t)\rangle_{eq} = 2\eta k_B T \int \frac{d\omega'}{2\pi} \langle \hat{\xi}(\omega')\hat{\xi}(\omega - \omega')\rangle \hat{\chi}(\omega')\hat{\chi}(\omega - \omega')$$
$$= 2\eta\, k_B T \delta(\omega) \int \frac{d\omega'}{2\pi} \hat{\chi}(\omega')\hat{\chi}(-\omega'). \tag{31}$$

As an equilibrium term, $\mathcal{F}_\omega \langle u^2(t)\rangle_{eq}$ gives a DC contribution ($\omega = 0$) to the dynamical correlation and does not prevent the system from reaching a steady state.

In contrast, the second term $\mathcal{F}_\omega \langle u^2(t)\rangle_{neq}$ in the right-hand side of Eq. (29) has a non-equilibrium origin. It disappears when the non-equilibrium force vanishes and is given by

$$\mathcal{F}_\omega \langle u^2(t)\rangle_{neq} = \int \frac{d\omega'}{2\pi} \hat{F}(\omega')\hat{F}(\omega - \omega')\hat{\chi}(\omega')\hat{\chi}(\omega - \omega'). \tag{32}$$

This term can be linked to the entropy spectral density $S_r(\omega,\omega')$, defined in Eq. (28). As a result, Eq. (32) can be written as follows

$$\mathcal{F}_\omega \langle u^2(t)\rangle_{neq} = T \int \frac{d\omega'}{2\pi} \frac{\hat{\chi}(\omega - \omega')}{(i\omega')} \hat{S}_r(\omega,\omega'), \tag{33}$$

which corresponds to Eq. (8) of the main text.

## Temperature dependence of the soft mode

The soft mode frequency and the damping or line width are strongly temperature dependent. We model the temperature dependence from

**Table 1 | Fitting parameters for the temperature dependence of the soft mode frequency $\omega_0$ and the damping $\eta$ for SrTiO$_3$ and KTaO$_3$**

|  |  | $a_0$ (THz) | $a_1$ (THz/K) | $a_2$ (THz/K$^2$) |
|---|---|---|---|---|
| SrTiO$_3$ | $\omega_0$ | 0.078 | 0.0137 | $-17 \times 10^{-6}$ |
|  | $\eta$ | 0.005 | 0.001 | $7 \times 10^{-6}$ |
| KTaO$_3$ | $\omega_0$ | 0.42 | 0.013 | $-18 \times 10^{-6}$ |
|  | $\eta$ | $-0.008$ | 0.0019 | $10 \times 10^{-6}$ |

data taken from Vogt[70] and fitting to a second-order polynomial,

$$x(T) = a_0 + a_1 T + a_2 T^2. \tag{34}$$

Here, $x = \omega_0, \eta$ is either the soft mode frequency $\omega_0$ or the damping $\eta$. In the past, other parametrizations of the soft mode have been proposed, e.g., the four-parameter model by Barrett[88]. However, for our purpose, a fit according to Eq. (34) provides a reasonable accuracy within the discussed temperature range. The fitting parameters are given in Table 1, while a comparison of the quadratic fit with experimental data is reported in Fig. 6 for SrTiO$_3$ (Fig. 6a) and KTaO$_3$ (Fig. 6b) materials, showing good agreement both for the soft mode frequency and damping.

## Coupled phonon modes

In the main text, we describe the driving of a single phonon mode. However, the strong-field excitation of phonons introduces the coupling with other phonon modes, as discussed in detail for the SrTiO$_3$ in ref. 22. Hence, one could wonder if this coupling leads to an additional source of entropy production. In the following, we will show that the entropy production of the medium is only due to an external driving field and not via the phonon-phonon coupling. Hence, off-resonant IR-active modes do not contribute to the total entropy production of the medium and, thus, to the absorbed heat.

We consider two modes denoted by $\hat{u}_0$ and $\hat{u}_1$, which are coupled by an interaction potential $V = V(\hat{u}_0, \hat{u}_1)$. The dynamics in frequency domain reads

$$(-\omega^2 + \omega_0^2 + i\omega\eta)\hat{u}_0(\omega) = \sqrt{2\eta T}\hat{\xi}_0(\omega) + \hat{F}(\omega) - \left[\frac{d}{du_0}V\right](\omega) \tag{35}$$

$$(-\omega^2 + \omega_1^2 + i\omega\eta)\hat{u}_1(\omega) = \sqrt{2\eta T}\hat{\xi}_1(\omega) + \hat{F}(\omega) - \left[\frac{d}{du_1}V\right](\omega) \tag{36}$$

Here, we do not specify the shape of $V = V(\hat{u}_0, \hat{u}_1)$ to ensure generality (typically $V$ is given as a polynomial in $\hat{u}_0, \hat{u}_1$).

Since the noise $\xi_0$ is independent of $\xi_1$, our approach of path integrals can be easily generalized to the present case, giving rise to the following expression for the total entropy production of the medium:

$$\dot{s}(t) = \int \frac{d\omega}{2\pi} e^{-i\omega t} \int \frac{d\omega'}{2\pi} \frac{1}{2T} \left[\hat{v}_0(\omega')F(\omega - \omega') + \hat{v}_0(\omega - \omega')F(\omega')\right]$$
$$+ \int \frac{d\omega}{2\pi} e^{-i\omega t} \int \frac{d\omega'}{2\pi} \frac{1}{2T} \left[\hat{v}_1(\omega')F(\omega - \omega') + \hat{v}_1(\omega - \omega')F(\omega')\right]. \tag{37}$$

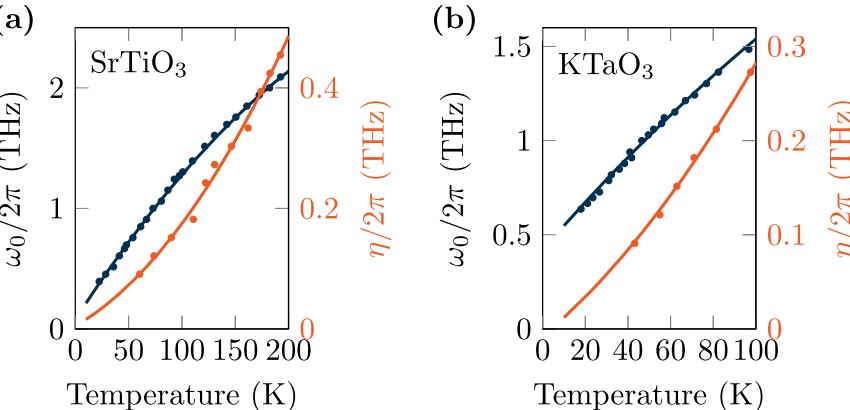

**(a)** SrTiO$_3$

**(b)** KTaO$_3$

**Fig. 6 | Temperature dependence of the soft mode.** Frequency (blue) and damping (orange) are plotted for **a** SrTiO$_3$ and **b** KTaO$_3$. Dots correspond to experimental data taken from Vogt[70]. Solid lines show the quadratic fit for comparison.

Indeed, the interaction term is due to a potential and therefore will produce only a boundary term in the expression for $\dot{s}(t)$. This can be seen easily in real space

$$\frac{\eta}{2T}\int dt\left(v_0\frac{d}{du_0}V(u_0,u_1)+v_1\frac{d}{du_1}V(u_0,u_1)\right)$$
$$=\frac{\eta}{2T}\int dt\frac{d}{dt}V(u_1,u_2). \tag{38}$$

Being expressed as a total time-derivative, this term does not contribute to the entropy production of the medium.

Equation (37) implies that contributions to the entropy production of the medium only arise when $\hat{F}$ and $\hat{v}_\alpha=i\omega\hat{u}_\alpha$, with $\alpha=0,1$, overlap. As a consequence, a silent mode does not significantly contribute to entropy production.

## Data availability
The presented data are available from the authors upon request.

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

## Acknowledgements

We thank Jérémy Vachier, Ivana Savic, Michael Fechner, and Michael Först for valuable insights and discussions. L.C. acknowledges support from the Alexander Von Humboldt foundation. H.L. acknowledges support by the Deutsche Forschungsgemeinschaft (DFG) through the SPP 2265 under the grant number LO 418/25-1. R.M.G. acknowledges support from the Swedish Research Council (VR starting grant No. 2022-03350) and Chalmers University of Technology. Computational resources were provided by the Swedish National Infrastructure for Computing (SNIC) via the National Supercomputer Centre (NSC).

## Author contributions

L.C., H.L., and R.M.G. contributed to the formalism, the manuscript, and interpretation of the results. Theoretical results were derived by L.C. and R.M.G. while numerical results were obtained by R.M.G.

## Funding

## Competing interests

The authors declare no competing interests.
