## [Peer Review File · Nature Communications]

REVIEWER COMMENTS

Reviewer #1 (Remarks to the Author):

The authors of this paper study entropy production in pumped-probed crystals where light excites the phonons of the crystal. The crystal is modeled as linearly coupled harmonic oscillators undergoing an underdamped isothermal Langevin dynamics. A time dependent driving describes the effect of the light driving the system out of equilibrium. Two theoretical results are put forward: First, entropy production is expressed and studied in Fourier representation (equation 4 and 5). Second, the power spectrum of the crystal can be expressed in terms of the frequency resolved entropy production (equation 6 to 8).

The authors then point out that these results can be used to calculate the frequency resolved entropy production from experiments and illustrate it using two applications.

Overall, the paper is clear and well written, but I fail to see good reasons to justify publication in a high impact journal such as Nature Communications.

Dissipation in driven harmonic systems has been extensively studied in the literature, e.g. H. C. Fogedby, A. Imparato J. Stat. Mech. (2012) P04005; N Freitas, JP Paz, Phys. Rev. E 90, 069903 (2014).

Spectrally decomposing entropy production is not a major achievement and has been previously considered in the context of parametrically driven harmonic systems, N Freitas, JC Delvenne, M Esposito, Phys. Rev. X 10, 031005 (2020), and linear response theory, D. Forastiere, R. Rao and M. Esposito, "Linear Stochastic Thermodynamics", New J. Phys. 24, 083021 (2022).

I also think that what the authors call entropy production is instead heat flow divided by temperature, see e.g. Martin Luc Rosinberg et al 2016 EPL 113 10007. The change in Shannon entropy of the system is missing to recover the full entropy production, see e.g. U. Seifert Phys. Rev. Lett. 95, 040602 (2005). To clarify this point, writing the energy and entropy balance explicitly would be helpful.

Pointing out that the theoretical results can be used to obtain the frequency resolved entropy production (or rather heat) in terms of experimentally accessible quantities is the most interesting aspect of the paper, but is not enough to justify publication in Nature Communications.

Minor comment:

There is a typo in (4): the second frequency should be primed.

Reviewer #2 (Remarks to the Author):

The authors present a study on entropy production in ultrafast phonon-pumping experiments. They develop a description of the entropy production rate and connect it to X-ray data from topical experiments on the quantum paraelectric SrTiO₃. The authors use a stochastic treatment of the phonon equations of motion, which is on the forefront of theoretical development in the field. In my opinion, the present study makes a significant contribution to bridging theory and experiment in the field of ultrafast dynamics, is relevant to a broad readership across condensed matter physics and nonlinear optics, and definitely deserves consideration for publication in Nature Communications.

There are a number of questions and comments though that I believe the authors should address before the manuscript can proceed towards publication. I will elaborate on the details below.

1) Clarifications

1.1) Abstract: What "striking properties" do the authors refer to? This is a bit general.

1.2) Abstract: The authors write that "(...) thermodynamic analysis under fast driving has never been connected to microscopic experiments (...)". This has to be rephrased in my opinion. As an example, in the field of ultrafast magnetism there is a long-standing debate about whether the inverse Faraday effect can be described within a thermodynamic effective magnetic field picture or requires a different treatment on femtosecond timescales, see, e.g., [Reid, Phys. Rev. B 81, 104404 (2010)], [Popova, Phys. Rev. B 84, 214421 (2011)], and [Gorelov, Phys. Rev. B 88, 220411(R) (2013)]. Also in the field of nonlinear phononics there is an ongoing debate about to which degree the adiabatic approximation is valid under coherent driving. I therefore think that this claim needs to be softened throughout the manuscript - the results are interesting enough.

1.3) Abstract: The authors write that their formalism provides "(...) particle-resolved insights into the dynamics of thermodynamic processes at ultrashort timescales." This is unclear to me, what particle and resolution is meant here? The formalism deals with collective excitations.

1.4) In the abstract, the authors write that the spectral entropy production / power spectrum of the displacement-displacement correlation show a broad peak beside the eigenmode resonance. In the discussion section it says that this peak appears besides twice the eigenfrequency of the soft mode.

In Fig. 2 c) and d), the peak looks much closer to ω than to 2ω . Can the authors clarify what they mean here?

1.5) Connected to the above point: according to Eq. (25), the spectral entropy production depends on the square of the driving force. If the driving force is resonant with the eigenfrequency (ω) of the soft mode, then the spectral entropy production should be at 2ω . Why is this not the case in Fig. 2 c) and d)?

2) Entropy production from nonlinear phonon coupling

2.1) Page 3: The authors write that the 5.19 THz mode "(...) is silent because there is no spectral overlap with the driving field (...)". I think the phrasing is misleading here, because the mode being silent is independent of the spectral range of the laser pulse.

2.2) Page 3, connected to the above point: The authors write that the entropy production for the coupled mode(s) is negligible, which seems in contrast to the experiment they refer to, which explicitly points out the importance of nonlinear phonon coupling. The 5.19 THz mode is not directly driven by the laser pulse, but indirectly through the resonantly excited soft mode. Since the driven phonon mode can be written formally in terms of the driving force of the laser (e.g. Eq. (23)), the coupled mode can be written in terms of the driving force of the laser as well, where the prefactor is the nonlinear susceptibility coming from the nonlinear phonon coupling. Would this give a notable contribution to the entropy production? It would be good to see a quantitative comparison.

3) Previous work

3.1) Page 3: The authors refer to previous X-ray measurements in SrTiO₃. In this context, a recent study could be taken into account: [Fechner, arXiv:2301.08703].

3.2) Eqs. (1) and (2): the authors write that in contrast to previous work, they take into account an uncorrelated noise term in the equations of motion to model thermal fluctuations. This has recently been done in another study: [Juraschek, Phys. Rev. Lett. 124, 117401 (2020)].

4) Temperature dependence

4.1) Fig. 3 and page 4: In the discussion of the temperature dependence of the entropy production rate, it would be good to see a graph actually showing σ vs temperature. The authors write that at very low temperatures <15 K, the absolute value of σ increases rapidly - this should be shown here, for example in a σ vs T graph.

4.2) Connected to the previous point: The authors write that at low temperatures, thermal noise is suppressed and nonequilibrium entropy production is enhanced. Does that mean an absolute enhancement or a relative enhancement (relative to the equilibrium contribution)? This connection isn't immediately clear to me right now.

4.3) Fig. 3: The temperature dependence for SrTiO₃ and KTaO₃ seem to be qualitatively different, e.g. a maximum of σ for SrTiO₃ and no maximum of σ for KTaO₃. Can this be traced back to the materials' properties as different types of paraelectrics?

5) Page 4: The authors connect the discussion of the temperature dependence of the entropy production rate to the concept of "entropons", which some of the authors introduced in another preprint, Ref. [67]. This needs to be elaborated on. What are entropons and how do they appear in the present study? Where does the "entropon broadening" come into play? The concept of entropons is referred to several times in manuscript and I think it is not enough referring to another preprint at this point.

6) Eq. (6): The power spectrum is given in terms of an equilibrium and a nonequilibrium contribution and is then connected to the entropy spectral density in Eq. (8). Does this include laser-induced heating? The coherently driven phonons will decay into the continuum of phonon modes and lead to a new equilibrium right after their lifetime. While this is often just a few Kelvin, it may become relevant at low temperatures.

7) General: What are the implications of being able to compute the entropy production rate? Can this information be used to learn something new about the properties of the materials?

Minor suggestions:

- In the main text, the entropy production rate is called σ , whereas in the methods it's called s . This could be unified.

- In some parts, a.m.u. is used for the atomic mass unit, and in others "u". This could be unified.

- Fig. 2: b) and c) could be switched to make the order easier for the eye. Also, I'd suggest to plot a) for the entire frequency range until 6 THz, in order to be able to compare it directly with the other three.

- Fig. 3: a) and b) could be made into two columns, then one could directly compare the respective graphs for SrTiO₃ and KTaO₃ next to each other.

- Fig. 4: The ensemble of solutions should be plotted more clearly. In printout, the individual graphs are barely visible.

Answering the questions suggested by Nature Communications:

- What are the noteworthy results?

The derivation of a theoretical formalism for entropy production in coherent phonon pumping experiments.

- Will the work be of significance to the field and related fields? How does it compare to the established literature? If the work is not original, please provide relevant references.

The work makes a significant contribution to the field of ultrafast dynamics by providing a new link between theory and experiment. Entropy production has so-far not been considered in coherent phonon pumping literature.

- Does the work support the conclusions and claims, or is additional evidence needed?

The theoretical and computational results are sound. The claim that thermodynamic analysis for ultrafast driving has never been connected to experiments should be rephrased, see my comment 1.2 above.

- Are there any flaws in the data analysis, interpretation and conclusions? - Do these prohibit publication or require revision?

No major flaws, for a detailed assessment please see the comments above.

- Is the methodology sound? Does the work meet the expected standards in your field?

Yes.

- Is there enough detail provided in the methods for the work to be reproduced?

Yes.

REPLY to Referee 1:

Referee: *The authors of this paper study entropy production in pumped-probed crystals where light excites the phonons of the crystal. The crystal is modeled as linearly coupled harmonic oscillators undergoing an underdamped isothermal Langevin dynamics. A time dependent driving describes the effect of the light driving the system out of equilibrium. Two theoretical results are put forward: First, entropy production is expressed and studied in Fourier representation (equation 4 and 5). Second, the power spectrum of the crystal can be expressed in terms of the frequency resolved entropy production (equation 6 to 8). The authors then point out that these results can be used to calculate the frequency resolved entropy production from experiments and illustrate it using two applications.*

Overall, the paper is clear and well written, but I fail to see good reasons to justify publication in a high impact journal such as Nature Communications.

Our reply: We appreciate the referee's review and recognition of our manuscript, despite their objection to publication. We fully understand the concerns that have been raised. However, it is crucial to highlight that our work represents a significant advancement in the field of solid-state physics, specifically in the realm of ultrafast dynamics of quantum materials.

While the formalism of entropy production in individual particles and small-scale (quantum) systems has been extensively explored within the framework of stochastic thermodynamics, our study delves into a distinct conceptual framework. Rather than focusing on single particles, we examine collective states of matter, known as quasiparticles, which play a pivotal role in the behavior of quantum materials. State-of-the-art experimental techniques allow for probing collective excitations in the far non-equilibrium regime with high precision and on ultrafast time-scales. In these systems, thermodynamics has not been established and understood on the same level, as e.g., in biophysical systems.

In pump-probe experiments, heat is commonly modeled using empirical heat equations that lack microscopic details of the material. In contrast, our work represents a significant advancement by providing a more detailed conceptual framework to address this issue. In particular, we derive the characteristic time-scale of entropy production, making it a central quantity in understanding current solid-state physics experiments. Here, a key example where our approach might be extremely powerful is the ultrafast switching of magnetization using laser pulses. In these experiments, ultrafast jumps of the magnetization can be probed. However, while there are various physical mechanisms in the manipulation of spin state, a local heating of a sample might lead to a similar magnetization drop [Nat. Comm., 3: 666 (2012)], which can (sometimes) only hardly be distinguished. Having control over the involved time scale of the heat production provides us with a novel method to discriminate competing effects. While SrTiO₃ itself is nonmagnetic, we note that it is currently very actively discussed with respect to transient magnetization due to the phonon inverse Faraday effect [arXiv:2210.01690] and represents a promising candidate for induced magnetic switching in layered structures involving ferromagnetic materials [arXiv:2305.11551].

In the present manuscript we showed for the first time how entropy production (closely related to heat production) can be extracted from observables (note that entropy itself has been known for almost two decades, but cannot be measured directly). We developed the example for the THz driving of soft modes of SrTiO₃ and KTaO₃ and signatures from time-resolved X-ray scattering.

Our study will stimulate future research activity:

- Probing phonon dynamics has emerged as a highly active research field, with various kinds of experimental probes. Here, we note that another interesting angle is the diffuse X-ray scatter-

ing which probes $\langle u(t) \rangle$ a signature with direct features of entropy production, according to our theory. Extending the scope of this work to further experiments promises to extract thermodynamic quantities naturally in the future and embed these results into the discussion of physical effects.

- Our approach can be generalized into the field of ultrafast thermodynamics, and extended towards other collective excitations, such as magnons.
- Also, we expect that the exciting application of stochastic thermodynamics to collective states of matter will in turn open new questions and stimulate further research in the field of non-equilibrium statistical physics. Here, our paper provides a novel direction in stochastic thermodynamics going beyond existing cases of interest, such as low-density soft matter systems.

Inspired by the comment of the referee, we extended our discussion section to lay out more clearly the potential of our results and also connect it stronger to the aforementioned example of ultrafast magnetic switching.

Referee: *Dissipation in driven harmonic systems has been extensively studied in the literature, e.g. H. C. Fogedby, A. Imparato J. Stat. Mech. (2012) P04005; N Freitas, JP Paz, Phys. Rev. E 90, 069903 (2014). Spectrally decomposing entropy production is not a major achievement and has been previously considered in the context of parametrically driven harmonic systems, N Freitas, JC Delvenne, M Esposito, Phys. Rev. X 10, 031005 (2020), and linear response theory, D. Forastiere, R. Rao and M. Esposito, "Linear Stochastic Thermodynamics", New J. Phys. 24, 083021 (2022).*

Our reply: We fully agree with the referee about prior research on driven harmonic systems, linear response, and linear stochastic thermodynamics. We are aware of these works and have now included references and a discussion in the new version of the manuscript.

However, it is important to highlight that the core focus of our manuscript extends beyond the mathematical formalism. Our primary contribution lies in the exploration of the immense potential for applying these tools within the dynamic and rapidly evolving research domain of ultrafast dynamics in quantum materials. Through our work, we present a novel perspective that showcases the exciting prospects arising from the use of these tools in solid-state physics.

The physical outcome of our investigation, which involves the meticulous tracking of entropy and heat production during the ultrafast excitation of phonons in SrTiO₃ and KTaO₃, represents a groundbreaking discovery. This result is particularly thrilling as it relies on fundamental observables that can be readily and routinely probed in state-of-the-art experiments. The ability to access such fundamental observables paves the way for deeper insights and opens new avenues of exploration within solid-state physics.

Furthermore, we hope that our work will foster collaboration and synergy between the communities of ultrafast science and stochastic thermodynamics. By bridging these two disciplines, we aim to establish a common framework that will not only enhance our understanding of ultrafast dynamics in quantum materials but also propose further applications in stochastic thermodynamics.

Referee: *I also think that what the authors call entropy production is instead heat flow divided by temperature, see e.g. Martin Luc Rosinberg et al 2016 EPL 113 10007. The change in Shannon entropy of the system is missing to recover the full entropy production, see e.g. U. Seifert Phys. Rev. Lett. 95, 040602 (2005). To clarify this point, writing the energy and entropy balance explicitly would be helpful.*

Our reply: We agree with the referee. In the new version of the paper, we have replaced “entropy

production” with the so-called “entropy production of the medium”, i.e. the ratio between heat and thermal temperature, and we have carefully commented on this point.

Once again, we stress that here we were able to relate this quantity to time-resolved X-ray scattering data, which allowed us to compute this observable explicitly for a phonon, i.e., a collective excitation.

Referee: *Pointing out that the theoretical results can be used to obtain the frequency resolved entropy production (or rather heat) in terms of experimentally accessible quantities is the most interesting aspect of the paper, but is not enough to justify publication in Nature Communications.*

Our reply: The core of our manuscript lies in the application of stochastic thermodynamics to laser-excited phonon modes, which represents a novel aspect in ultrafast materials dynamics. Therefore, we convinced that the implications of our findings fully justify their publication in Nature Communications. Our work constitutes a notable advancement in ultrafast solid-state physics, presenting a microscopic theory that elucidates the heat and entropy production mediated by phonons in highly non-equilibrium regimes. Our paper demonstrates, for the first time, that conventional observables together with the laser source can provide all the necessary information to reconstruct entropy and heat production.

Moreover, our approach forges a valuable connection between two distinct realms of research: ultrafast dynamics and stochastic thermodynamics. While stochastic thermodynamics has predominantly focused on the calculation of heat and entropy production in dilute systems, including gases, low-density colloids, or bacteria, we extend this formalism to quasi-particles, i.e., collective excitations of the solid state. To the best of our knowledge, this particular aspect has remained unexplored within the existing literature. Consequently, our work possesses the potential to exert a profound impact on the field of solid-state physics, spurring the development of innovative measurement techniques inspired by stochastic thermodynamics.

Referee: *Minor comment: There is a typo in (4): the second frequency should be primed.*

Our reply: We thank the referee for the careful reading. We have corrected this typo in the new version of the paper.

REPLY to Referee 2:

Referee: *The authors present a study on entropy production in ultrafast phonon-pumping experiments. They develop a description of the entropy production rate and connect it to X-ray data from topical experiments on the quantum paraelectric SrTiO₃. The authors use a stochastic treatment of the phonon equations of motion, which is on the forefront of theoretical development in the field. In my opinion, the present study makes a significant contribution to bridging theory and experiment in the field of ultrafast dynamics, is relevant to a broad readership across condensed matter physics and nonlinear optics, and definitely deserves consideration for publication in Nature Communications.*

There are a number of questions and comments though that I believe the authors should address before the manuscript can proceed towards publication. I will elaborate on the details below.

Our reply: We are grateful to the referee for taking time and reviewing our manuscript. Furthermore, we are excited to read the recommendation to consider our manuscript for publication after revision. Following the comments and suggestions, we have made several improvements to our manuscript. Our replies to the points raised by the referee are given below.

1. **Referee:** *Clarifications:*

1.1) **Referee:** *Abstract: What "striking properties" do the authors refer to? This is a bit general.*

Our reply: We agree with the referee. To give a few examples that have been recently and widely discussed in the literature, we now mention *transient superconductivity and ferroelectricity, ultrafast magnetization and demagnetization, and Floquet engineering.*

1.2) **Referee:** *Abstract: The authors write that "(...) thermodynamic analysis under fast driving has never been connected to microscopic experiments (...)". This has to be rephrased in my opinion. As an example, in the field of ultrafast magnetism there is a long-standing debate about whether the inverse Faraday effect can be described within a thermodynamic effective magnetic field picture or requires a different treatment on femtosecond timescales, see, e.g., [Reid, Phys. Rev. B 81, 104404 (2010)], [Popova, Phys. Rev. B 84, 214421 (2011)], and [Gorelov, Phys. Rev. B 88, 220411(R) (2013)]. Also in the field of nonlinear phononics there is an ongoing debate about to which degree the adiabatic approximation is valid under coherent driving. I therefore think that this claim needs to be softened throughout the manuscript - the results are interesting enough.*

Our reply: We are grateful to the referee for highlighting the interesting debate surrounding the inverse Faraday effect. In fact, we had already planned to extend our discussion section to include magnetization and demagnetization effects. Moreover, our microscopic approach to heat production by phonons may offer valuable insights into the investigation of ultrafast demagnetization through thermal heat. Consequently, we have incorporated the referee's proposed references and supplemented them with several more recent papers that are pertinent to this discussion.

Also, we have rephrased our statement in the abstract, and removed "thermodynamic analysis under fast driving has never been connected to microscopic experiment" as we believe the second part of the sentence, i.e., "The characterization of the ultrafast thermodynamic properties within the material is key for their control and design." is a valid and fitting statement by itself.

1.3) **Referee:** *Abstract: The authors write that their formalism provides "(...) particle-resolved insights into the dynamics of thermodynamic processes at ultrashort timescales." This is*

unclear to me, what particle and resolution is meant here? The formalism deals with collective excitations.

Our reply: We agree with the referee and rephrased this sentence in the new version of the paper: "Here, we develop the ultrafast stochastic thermodynamics for laser-excited phonons at ultrafast timescales."

- 1.4) **Referee:** *In the abstract, the authors write that the spectral entropy production / power spectrum of the displacement-displacement correlation show a broad peak beside the eigenmode resonance. In the discussion section it says that this peak appears besides twice the eigenfrequency of the soft mode. In Fig. 2 c) and d), the peak looks much closer to omega than to 2omega. Can the authors clarify what they mean here?*

Our reply: We thank the referee for the insightful question. Before commenting on Figure 2, we would like to discuss a related feature in Figure 3. In Figure 3, we simulate the phonon dynamics and entropy production for a narrow pulse with driving frequency $\omega_d = 0.75$ THz. Following the experiment of Vogt [PRB, 51, 8046, 1995], the ferroelectric softmode of SrTiO₃ should be in resonance at a temperature of ≈ 50 K. If you now compare the absolute of the entropy production as well as the Fourier transform of the power spectrum in Figure 3(a) for a temperature of 50 K, you see a single symmetric peak. The peak broaden to the right for temperatures below 50 K and to the left for temperatures above 50 K.

In contrast, in Figure 2(c) and 2(d), we compute the spectral entropy production and power spectrum for a THz pulse constructed with 2 Gaussians at driving frequencies $\omega_{d,1} = 0.75$ THz and the higher harmonic $\omega_{d,2} = 1.5$ THz. This combination seems to reproduce the experimental profile sufficiently. The simulation / experiment is performed at a temperature of $T = 100$ K, where the soft-mode frequency is much higher as compared to $\omega_{d,1}$, i.e., $\omega_0(100 \text{ K}) \approx 1.66$ THz. If you compare the spectrum of the laser pulse with the spectrum of the phonon mode, i.e., Figure 2(a), and 2(b), they overlap strongly with frequencies below ≈ 2.5 THz. Therefore, we expect that the featureless and symmetric form of the entropy production (Figure 2(c)) and power spectrum (Figure 2(d)) are strongly connected with this spectral overlap.

- 1.5) **Referee:** *Connected to the above point: according to Eq. (25), the spectral entropy production depends on the square of the driving force. If the driving force is resonant with the eigenfrequency (omega) of the soft mode, then the spectral entropy production should be at 2omega. Why is this not the case in Fig. 2 c) and d)?*

Our reply: According to equation 25, the spectral entropy production is given by the convolution $\hat{s}(\omega) = \hat{F}(\omega) \star \left(\hat{F}(\omega) \hat{\chi}(\omega) \right)$. As the referee points out, for a resonance with the phonon mode, we expect the spectral entropy production to occur at twice the driving or phonon mode frequency. However, in the experiment / simulation in Figure 2, the soft mode is not in resonance with the driving frequency, which is significantly lower. Yet due to the strong field, the ionic motion is induced with a fairly low, off-resonant component. As a result, we expect the convolution with the susceptibility to shift the entropy production to lower frequencies than $2\omega_0$.

2. **Referee:** *Entropy production from nonlinear phonon coupling*

- 2.1) **Referee:** *Page 3: The authors write that the 5.19 THz mode "(...) is silent because there is no spectral overlap with the driving field (...)". I think the phrasing is misleading here, because the mode being silent is independent of the spectral range of the laser pulse.*

Our reply: Thank you for pointing out. Of course, the 5.15 THz mode is infrared active, i.e., not silent. The silent mode would come afterwards at ≈ 7.9 THz. Here, we mean

there is no spectral overlap and as a result no direct excitation of the 5.15 THz mode. We corrected the statement.

- 2.2) **Referee:** *Page 3, connected to the above point: The authors write that the entropy production for the coupled mode(s) is negligible, which seems in contrast to the experiment they refer to, which explicitly points out the importance of nonlinear phonon coupling. The 5.19 THz mode is not directly driven by the laser pulse, but indirectly through the resonantly excited soft mode. Since the driven phonon mode can be written formally in terms of the driving force of the laser (e.g. Eq. (23)), the coupled mode can be written in terms of the driving force of the laser as well, where the prefactor is the nonlinear susceptibility coming from the nonlinear phonon coupling. Would this give a notable contribution to the entropy production? It would be good to see a quantitative comparison.*

Our reply: We thank the referee for raising this point. We have carefully thought about this issue while working on our manuscript. However, we missed to provide a more detailed discussion. In the present version, we show analytically that a phonon mode with vanishing spectral overlap with the driving force does not contribute to entropy production (see the new subsection on "coupled phonon modes" in the revised method section).

For completeness we also provide the argument below. Let us assume that our crystal can be described by the dynamics of two phonon modes, say \hat{u}_0 and \hat{u}_1 , with the following expression in Fourier space:

$$(-\omega^2 + \omega_0^2 + i\omega\eta)\hat{u}_0(\omega) = \sqrt{2\eta T}\hat{x}_0(\omega) + \hat{F}(\omega) - \left[\frac{d}{du_0} V(\hat{u}_0, \hat{u}_1) \right] (\omega) \quad (1)$$

$$(-\omega^2 + \omega_1^2 + i\omega\eta)\hat{u}_1(\omega) = \sqrt{2\eta T}\hat{x}_1(\omega) + \hat{F}(\omega) - \left[\frac{d}{du_1} V(\hat{u}_0, \hat{u}_1) \right] (\omega). \quad (2)$$

$V(\hat{u}_0, \hat{u}_1)$ is a general interacting potential between the phonon modes (usually a polynomial).

Our approach of path integrals can be generalized to the present case, giving rise to the following expression:

$$\begin{aligned} \dot{s}(t) = & \int \frac{d\omega}{2\pi} e^{-i\omega t} \int \frac{d\omega'}{2\pi} \frac{1}{2T} [\hat{v}_0(\omega')F(\omega - \omega') + \hat{v}_0(\omega - \omega')F(\omega')] \\ & + \int \frac{d\omega}{2\pi} e^{-i\omega t} \int \frac{d\omega'}{2\pi} \frac{1}{2T} [\hat{v}_1(\omega')F(\omega - \omega') + \hat{v}_1(\omega - \omega')F(\omega')] \end{aligned} \quad (3)$$

This is because the interaction term is due to a potential and therefore will produce only an irrelevant boundary term in the expression for the entropy production rate. This can be verified in real space,

$$\frac{\eta}{2T} \int dt \left(v_0 \frac{d}{du_0} V(u_0, u_1) + v_1 \frac{d}{du_1} V(u_0, u_1) \right) = \frac{\eta}{2T} \int dt \frac{d}{dt} V(u_1, u_2). \quad (4)$$

Being expressed as a total time-derivative, this term does not significantly contribute to the entropy production.

Hence, equation (3) implies that contributions to entropy production involve the product of \hat{F} and $\hat{v}_\alpha = i\omega\hat{u}_\alpha$, with $\alpha = 0, 1$. However, if \hat{u}_1 is not excited, its contribution to the total entropy production vanishes.

3. Referee: *Previous work*

- 3.1) **Referee:** *Page 3: The authors refer to previous X-ray measurements in SrTiO₃. In this context, a recent study could be taken into account: [Fechner, arXiv:2301.08703].*

Our reply: We thank the referee for bringing up this reference. In fact, this work has appeared on arXiv shortly before we finalized our first draft and we have missed it. It is exciting to see how Fechner *et al.* derive the spectra for $\langle u(t)^2 \rangle$ using time-resolved diffuse X-ray scattering. Even though, we cannot directly use these results in the present manuscript, we provide the reference in connection to equation (8) in the main text. Additionally, we added a general reference for diffuse X-ray scattering [*Nature Physics* **9**, 790–794 (2013)].

- 3.2) **Referee:** *Eqs. (1) and (2): the authors write that in contrast to previous work, they take into account an uncorrelated noise term in the equations of motion to model thermal fluctuations. This has recently been done in another study: [Juraschek, Phys. Rev. Lett. 124, 117401 (2020)].*

Our reply: We thank the referee for suggesting this paper. We added this reference together with [*J. Stat. Mech.* (2007) P09018] to the paragraph between equations (1) and (2).

4. **Referee:** *Temperature dependence*

- 4.1) **Referee:** *Fig. 3 and page 4: In the discussion of the temperature dependence of the entropy production rate, it would be good to see a graph actually showing sigma vs temperature. The authors write that at very low temperatures < 15 K, the absolute value of sigma increases rapidly - this should be shown here, for example in a sigma vs T graph.*

Our reply: We thank the referee for the great point. We have revised the paragraph and came to the conclusion to add a plot for the entropy production, according to equation (1), at times where the laser has fully decayed. This plot highlights the relationship between entropy production and temperature. Notably, our analysis indicates that the peak entropy production aligns with temperatures where the soft mode frequency is in resonance with the laser pulse. Our updated interpretation provides more clarity than our earlier explanation, which only focused on the increase in the peak spectral entropy production, corresponding to entropy production rate oscillations.

- 4.2) **Referee:** *Connected to the previous point: The authors write that at low temperatures, thermal noise is suppressed and nonequilibrium entropy production is enhanced. Does that mean an absolute enhancement or a relative enhancement (relative to the equilibrium contribution)? This connection isn't immediately clear to me right now.*

Our reply: Following the previous point, we removed this statement from the current manuscript. The relationship between entropy production and temperature should now be clear from the new Figure 4.

- 4.3) **Referee:** *Fig. 3: The temperature dependence for SrTiO₃ and KTaO₃ seem to be qualitatively different, e.g. a maximum of sigma for SrTiO₃ and no maximum of sigma for KTaO₃. Can this be traced back to the materials' properties as different types of paraelectrics?*

Our reply: We believe this assumption is correct. SrTiO₃ is a quantum paraelectric, where the ferroelectric soft mode becomes zero at finite temperatures. Yet, a ferroelectric phase transition is suppressed due to quantum fluctuations. In contrast, KTaO₃ remains cubic down to zero Kelvin, even though the softening of the ferroelectric mode can clearly be seen. As a result, the soft mode resonance with the laser pulse occurs at much higher temperatures for SrTiO₃, as compared to KTaO₃. At the same time, following the data of Vogt [*PRB* **51**, 8046 (1995)], the damping is qualitatively similar and much smaller for the relevant temperature range for KTaO₃, as compared to SrTiO₃.

5. **Referee:** *Page 4: The authors connect the discussion of the temperature dependence of the entropy production rate to the concept of "entropions", which some of the authors introduced in another preprint, Ref. [67]. This needs to be elaborated on. What are entropions and how do they appear in the present study? Where does the "entropion broadening" come into play? The concept of entropions is referred to several times in manuscript and I think it is not enough referring to another preprint at this point.*

Our reply: We agree with the referee, the reference to entropions is not obvious from our manuscript. We added a few lines on entropions to the respective paragraph in our text. Generally, entropions are novel collective excitations in active crystals, consisting of particles intrinsically out of equilibrium. In contrast to here, entropions are generated when each basic constituent of the crystal is internally out of equilibrium, while here the crystal is globally pushed out of equilibrium by the laser pulse. While an experimental realization of entropions is still unknown (to the best of our knowledge), candidates to observe entropions are available in soft (active) matter, for instance dense systems of bacteria, cells, and Janus colloids, internally driven by self-propulsion mechanisms.

6. **Referee:** *Eq. (6): The power spectrum is given in terms of an equilibrium and a nonequilibrium contribution and is then connected to the entropy spectral density in Eq. (8). Does this include laser-induced heating? The coherently driven phonons will decay into the continuum of phonon modes and lead to a new equilibrium right after their lifetime. While this is often just a few Kelvin, it may become relevant at low temperatures.*

Our reply: We do believe that there is a strong connection between our formalism and laser induced heating. It is important to mention that our formalism introduced in the present manuscript only takes into account phononic degrees of freedom, requiring a direct excitation due to a THz laser light. Decay effects from e.g. electronic or magnonic degrees of freedom into the phonon bath are not considered here. Vice versa, the interaction of the considered infrared active phonon mode with other degrees of freedom or other phonon modes is simplified by the stochastic force. Yet, we believe that laser induced heating would be an exciting application for future work. Therefore we added a few lines concerning laser induced heating and magnetization control to our discussion section.

7. **Referee:** *General: What are the implications of being able to compute the entropy production rate? Can this information be used to learn something new about the properties of the materials?*

Our reply: We thank the referee for the interest in the concept and would like to provide a brief outline. First, we mention that the formalism of entropy production in individual particles and small-scale (quantum) systems has been extensively explored within the framework of stochastic thermodynamics. Here, we propose a new angle by considering collective excitations. Hence, we build a connection between the communities of stochastic thermodynamics (soft matter) and ultrafast dynamics in solid state physics.

Second, we would like to point out that heat (in pump-probe experiments) is commonly modeled using empirical heat equations. Such equations often lack microscopic details of the material, and we hope to shed light into this mechanism. In fact, our formalism has full temporal resolution, allowing us to track the microscopic dynamics of heat and entropy production. An interesting example for our approach might be the laser-driven ultrafast switching of magnetization. Furthermore, there might be an interesting connection between the thermal entropy production and the production of information entropy. We currently see an increasing interest in using quantum materials for applications in information technology, which often require a fast control.

Referee: *Below, minor points*

1. **Referee:** *In the main text, the entropy production rate is called sigma, whereas in the methods it's called s. This could be unified.*

Our reply: we thank the referee for the careful reading. In the new version of the paper, we have uniformed the notation by using $\dot{s}(t)$ to denote the entropy production rate.

2. **Referee:** *- In some parts, a.m.u. is used for the atomic mass unit, and in others "u". This could be unified.*

Our reply: We thank the referee for the careful reading. In the new version of the paper, we have unified the notation, replacing "u" with "a.m.u."

3. **Referee:** *Fig. 2: b) and c) could be switched to make the order easier for the eye. Also, I'd suggest to plot a) for the entire frequency range until 6 THz, in order to be able to compare it directly with the other three.*

Our reply: We thank the referee for this suggestion. We now plot Fig. 2 (a), i.e., the spectrum of the laser pulse in the entire frequency range up to 6 THz. However, we would like to keep (b) and (c) as they are right now, as we believe (a) and (b) as well as (c) and (d) should be plotted on top of each other.

4. **Referee:** *Fig. 3: a) and b) could be made into two columns, then one could directly compare the respective graphs for SrTiO₃ and KTaO₃ next to each other.*

Our reply: We thank the referee for the suggestion. We now plot the results for SrTiO₃ and KTaO₃ in Figure 3 next to each other.

5. **Referee:** *- Fig. 4: The ensemble of solutions should be plotted more clearly. In printout, the individual graphs are barely visible.*

Our reply: We have modified Fig.4 to increase its readability.

REVIEWERS' COMMENTS

Reviewer #1 (Remarks to the Author):

The authors agreed with my comments and don't claim anymore that representing entropy production in Fourier space in a key achievement.

In their reply they state that "...the core focus of our manuscript extends beyond the mathematical formalism." I agree. I think that they mostly apply the well established framework of stochastic thermodynamics for driven harmonic systems to analyze ultrafast phonon-pumping experiments. But the paper does not really reflect this. For instance the abstract still claims "Here, we develop the ultrafast stochastic thermodynamics for laser-excited phonons at ultrafast timescales.".

In the reply the authors distinguish between conventional stochastic thermodynamics for single particle and their novel approach that deals with collective excitations. I find this distinction rather trivial for harmonic systems as collective excitations are simply normal modes.

They state in their reply that "Our primary contribution lies in the exploration of the immense potential for applying these tools within the dynamic and rapidly evolving research domain of ultrafast dynamics in quantum materials." I agree that this is their main contribution, but I don't know how immense is the potential for applying these tools which are limited to harmonic systems.

As I stated in my first report, the paper is sound and has value, mostly because it bring concepts from stochastic thermodynamics to a new field that is not used to it and shows that meaningfully and experimentally relevant predictions can be made. The theoretical and conceptual innovation from the standpoint of stochastic thermodynamics is however marginal. This is why I remain skeptical about the relevance of the paper for Nature Communications.

Reviewer #2 (Remarks to the Author):

The authors have responded to the raised issues in detail and made appropriate changes to the revised manuscript. In particular, the addition of the coupled phonons discussion in the methods section is very much appreciated.

I have a few more remaining points that I think could be addressed, after which I would recommend proceeding towards publication.

1) Authors' response to point 1.4: As I understand, the laser pulse is first modeled as consisting of two center frequencies of 0.75 and 1.5 THz. Has this been done to model the experimental pulse spectrum for Fig. 2 specifically? Later-on in the manuscript, only a pulse with one center frequency is used, is that right? Otherwise, the temperature dependence in Fig. 4 should hit two resonances corresponding to the two center frequencies, I think.

2) Fig. 4 is appreciated, but possibly an additional way of plotting the temperature dependence may tell more about the system. Right now, changing the temperature tunes the frequency of the phonon mode in and out of resonance with the laser pulse with a fixed center frequency. Accordingly, the entropy production peaks at resonance conditions, because the phonon gets excited most strongly there, but this is kind of expectable. The following scenario could tell more about the system itself: For every temperature, excite the phonon mode resonantly with constant intensity, by adjusting the center frequency, FWHM, and peak electric field of the pulse accordingly. This means that at every temperature, the phonon gets excited equally strongly, but now the different temperature dependences for STO and KTO may become apparent, such as the fact that in Figs. 3(a) and (b), STO has a peak of σ in temperature, whereas KTO does not.

3) A few typos: 'timescles' in the abstract and 'environmental' on page 2.

Reviewer #3 (Remarks to the Author):

The authors present theoretical models to demonstrate how it is possible to quantify entropy production in crystals from ultrafast pump-probe spectroscopic experiments. From the correspondance between the authors and the referees, I conclude that all points by referee 2 were addressed satisfactorily. The discussion with referee 1 appears to be more involved. Referee 1 has commanding knowledge of the field and the modifications proposed to the bibliography and to the text in general have greatly improved the manuscript. As far as I understand, all questions have been addressed properly.

Yet, the referee's main criticism remains: "Pointing out that the theoretical results can be used to obtain the frequency resolved entropy production (or rather heat) in terms of experimentally

accessible quantities is the most interesting aspect of the paper, but is not enough to justify publication in Nature Communications." On the one hand, the proposed connection with the experimental quantity is relatively thin. On the other hand, it provides a simple way of extracting invaluable physical information from a complex physical system. From my own biased perspective, I believe this simplicity together with the thorough theoretical treatment presented in the manuscript justify publication in Nature Communications.

REPLY to Referee 1:

Referee: *As I stated in my first report, the paper is sound and has value, mostly because it brings concepts from stochastic thermodynamics to a new field that is not used to it and shows that meaningfully and experimentally relevant predictions can be made. The theoretical and conceptual innovation from the standpoint of stochastic thermodynamics is however marginal. This is why I remain skeptical about the relevance of the paper for Nature Communications.*

Our reply: We sincerely appreciate the referee's efforts in providing a second report. We are pleased that the referee recognized the potential of our manuscript within the ultrafast dynamics community. Stochastic thermodynamics techniques offer an intriguing platform for investigating non-equilibrium dynamics in laser-driven quantum materials. Our theoretical and conceptual innovation can indeed in principle be extended beyond harmonic theory, which is a challenging task for the future.

REPLY to Referee 2:

Referee: *The authors have responded to the raised issues in detail and made appropriate changes to the revised manuscript. In particular, the addition of the coupled phonons discussion in the methods section is very much appreciated.*

I have a few more remaining points that I think could be addressed, after which I would recommend proceeding towards publication.

Our reply: We value the referee's efforts in offering a second report and additional suggestions for improving the manuscript. Further details are provided below.

Referee: *1) Authors' response to point 1.4: As I understand, the laser pulse is first modeled as consisting of two center frequencies of 0.75 and 1.5 THz. Has this been done to model the experimental pulse spectrum for Fig. 2 specifically? Later-on in the manuscript, only a pulse with one center frequency is used, is that right? Otherwise, the temperature dependence in Fig. 4 should hit two resonances corresponding to the two center frequencies, I think.*

Our reply: The referee is right. In the first part of the paper, we model the pump laser by a superposition of two Gaussians with frequencies $\omega_d = 0.75$ THz and $2\omega_d = 1.5$ THz. This form matches sufficiently well the experimental pump laser from Ref. [Nat. Phys. **15**, 387 (2019)], after fitting the appropriate intensity ratio of both Gaussians. This choice allowed us to connect closer to the X-ray diffraction data from [Nat. Phys. **15**, 387 (2019)] and compute the entropy production in a realistic framework. However, as we progress further in the paper, we delve deeper into the theoretical aspects of ultrafast entropy production and choose a simple description with a single Gaussian pulse for better accessibility. In the new version of the paper, we have explicitly commented on this point.

Referee: *2) Fig. 4 is appreciated, but possibly an additional way of plotting the temperature dependence may tell more about the system. Right now, changing the temperature tunes the frequency of the phonon mode in and out of resonance with the laser pulse with a fixed center frequency. Accordingly, the entropy production peaks at resonance conditions, because the phonon gets excited most strongly there, but this is kind of expectable. The following scenario could tell more about the system itself: For every temperature, excite the phonon mode resonantly with constant intensity, by adjusting the center frequency, FWHM, and peak electric field of the pulse accordingly. This means that at every temperature, the phonon gets excited equally strongly, but now the different temperature dependences for STO and KTO may become apparent, such as the fact that in Figs. 3(a) and (b), STO has a peak of sigma in temperature, whereas KTO does not.*

Our reply: This is an interesting remark by the referee which deepens the principle understanding of the data. We agree and, thus, we have provided a new figure (Figure 4b) in the new version of the paper. The new plot shows the total entropy production as a function of temperature after exciting the phonon mode resonantly with constant intensity for both SrTiO₃ and KTaO₃, as suggested by the referee. The computed total entropy production of the medium in resonant conditions increases with decreasing temperature and, in particular, diverges for vanishing temperature. This implies that, in resonant conditions, the smaller the temperature the larger the heat absorbed by the system.

Referee: *3) A few typos: 'timescles' in the abstract and 'environmental' on page 2.*

Our reply: We further thank the referee for carefully reading the manuscript. We have corrected both typos in the new version of the paper.

REPLY to Referee 3:

Referee: *The authors present theoretical models to demonstrate how it is possible to quantify entropy production in crystals from ultrafast pump-probe spectroscopic experiments. From the correspondence between the authors and the referees, I conclude that all points by referee 2 were addressed satisfactorily. The discussion with referee 1 appears to be more involved. Referee 1 has commanding knowledge of the field and the modifications proposed to the bibliography and to the text in general have greatly improved the manuscript. As far as I understand, all questions have been addressed properly.*

Yet, the referee's main criticism remains: "Pointing out that the theoretical results can be used to obtain the frequency resolved entropy production (or rather heat) in terms of experimentally accessible quantities is the most interesting aspect of the paper, but is not enough to justify publication in Nature Communications." On the one hand, the proposed connection with the experimental quantity is relatively thin. On the other hand, it provides a simple way of extracting invaluable physical information from a complex physical system. From my own biased perspective, I believe this simplicity together with the thorough theoretical treatment presented in the manuscript justify publication in Nature Communications.

Our reply: We appreciate the referee's thorough review of both the paper and the extended correspondence with the other two referees. We are pleased with the very positive evaluation of our manuscript and for suggesting it for publication in Nature Communications.